# Research on Impact of IoT on Warehouse Management

**DOI:** 10.3390/s23042213

**Published:** 2023-02-16

**Authors:** Aldona Jarašūnienė, Kristina Čižiūnienė, Audrius Čereška

**Affiliations:** 1Department of Logistics and Transport Management, Vilnius Gediminas Technical University, Plytinės Str. 27, LT-10105 Vilnius, Lithuania; 2Department of Mechanical and Material Engineering, Vilnius Gediminas Technical University, Plytinės Str. 25, LT-10105 Vilnius, Lithuania

**Keywords:** Internet of Things (IoT), warehouse, management, Industry 4.0

## Abstract

Automation and digitisation are the driving force of the Industrial Revolution 4.0. Industrial revolutions led to the mass production of goods, which increased the need for modern warehouses. Every year, the operation of warehouses becomes increasingly more complicated due to the increasing abundance of goods, thus the usual warehouse management strategies are no longer suitable. In order to cope with huge product flows, modern innovations should be used more extensively to manage these processes. Successful management will help provide quality service to rapidly changing business sectors. The Internet of Things (IoT) is a technology designed to process large amounts of data with maximum efficiency in real time. This technology can facilitate the implementation of smart identification, tracking, tracing, and management using radio frequency identification (RFID), infrared sensors, global positioning systems (GPS), laser scanners, and other detection tools. Such innovations as IoT have made a significant impact on warehousing operations. The aim of IoT is to perform administrative work, i.e., to efficiently manage warehouse data. IoT can be used to monitor and track goods, forecast demand trends, manage inventory, and perform other warehouse operations in real time. The key elements of a warehouse are sales and customer satisfaction. Implementing IoT improves financial performance, work productivity, and customer satisfaction. However, innovation requires additional investment in, for instance, implementation and maintenance. It is necessary to investigate how warehouse elements such as inventory accuracy or order processing time are affected by the internet of things in companies of different sizes. Research on the impact of IoT on warehouse management focuses on IoT advantages, disadvantages, mitigation risks, and the use of IoT in warehouses. The aim of this work is to research the impact of IoT on warehouse management in companies of different sizes and to determine whether the costs and benefits of IoT differ in the same scenario. As a result, the conceptual model for the adoption of IoT measures in warehouse companies was created, and its suitability was assessed by experts.

## 1. Introduction

Extensive production of goods launched by the industrial revolution intensifies the necessity for modern warehouses [1]. In recent years, the functioning of warehousing has gotten more unpredictable with an increasing number of things to be handled in a warehouse, thus customary and manual strategies for warehouse management are no longer required to deal with such an extent of activities. This prompted a more intensive use of modern innovation to manage these challenges [2].

The German expression “Industrie 4.0” was first introduced at the 2011 Hannover Messe trade fair. It incorporates the entire series of inventive operations and advancements to supplement new ideas within industry norms in the assembling area to serve rapidly advancing business sectors. In this context, these innovations have a positive impact on warehousing operations. IoT is one of the key examples [3]. The presence of IoT in a warehouse means human-to-machine and machine-to-machine interaction. It is the key technology for processing large amounts of data with the highest efficiency in a short time at the warehouse in real-time. It is an idea of a network, which can easily implement intelligent identification, location, monitoring, tracking, and control through radio frequency identification (RFID), infrared sensors, worldwide positioning frameworks (GPS), laser scanners, and other localisation gadgets. IoT helps to manage the administrative workload, making it easier to oversee and control warehouse data [4]. For instance, IoT can be used in warehouse management for monitoring, tracing goods, forecasting demand trends, inventory management, and other warehouse operations in real-time [5]. The main function of a warehouse is ensuring sales and customer satisfaction. The implementation of IoT improves financial performance, labour productivity, and customer satisfaction [6].

Research problem and novelty: The implementation and management of innovations used in warehousing processes provide conditions for obtaining information about the current situation at the desired time, which allows foreseeable solutions for improving operations or processes, for example, to organise the work of warehousing, transportation, and supply chains more efficiently. Thus, the Internet of Things (IoT) can indirectly or directly affect the business, the market in which it operates, even the entire sector, since the Internet of Things (IoT) is still a relatively new technological entity that promotes business development, greater profitability and competitiveness of the company, innovative companies, and the image of providing warehousing services. So far, there is a lack of research on the impact of IoT on warehousing operations, as well as the impact of IoT on warehouse management in companies of different sizes. The concept of the Industrial Revolution 4.0, and IoT in particular, is about the automation of industries in general and warehouses in particular. The implementation of innovations supporting the complexity of the functioning of warehouses requires time. However, flexibility and innovation come at additional costs and pose such challenges, as extensive investments are required for implementation, maintenance costs, shortage of skilled IT professionals, impacts on nature, and the structure of operations of warehouses. The impact of the implementation of IoT on such elements of warehousing processes as inventory accuracy or order lead time should be analysed in companies of different sizes.

Prior research of impacts of IoT on warehouse management predominantly focus on the advantages, disadvantages, mitigation risks, and use of IoT in warehouses. Therefore, the motivation behind this research is to examine the significant impact of IoT on warehouse management in companies of different sizes and to determine if IoT has different costs and benefits for the same scenario.

The objective of the research is to conduct a study of the impact of IoT on warehouse management based on company size.

Presentation of the research process model: In this study, the relevant literature was reviewed to analyse the concept of the importance of a warehouse, the functioning of a warehouse, and logistics—the elements that affect the performance of warehouses. The review of the relevant literature presented in this research was used to examine the application of IoT in warehouse management, its importance, and the impacts of IoT on warehouse management. The aim of the literature review was to identify the key performance indicators of a warehouse and possible costs and benefits of IoT that affect warehouse management. The outcome of this study will contribute to proposing a conceptual model for IoT adoption for warehousing companies. This research uses a qualitative empirical research approach, conducting semi-structured interviews to collect data from experts. Quantitative descriptive data analyses will be conducted to obtain results.

## 2. Methods and Methodology

### 2.1. Analysis of Scientific Literature on the Impact of IoT on Warehouse Management

In today’s modern era, more and more researchers and scientists invest their time in boosting warehousing by automating it and making it more efficient. Warehouse processes include storing, distributing, and controlling the flow of commodities within a warehouse [7]. Warehouses have always been a crucial hub in the flow of goods within a supply chain, covering various operations such as inventory, delivery, reception of goods, docking, etc. The work by [8] describes warehousing management as storage, sorting, dispatching of goods, and inventory forecasting. In this new advanced era, warehouses are of specific importance in interconnecting production and supply. In their analysis, Krauth et al. [9] inspect the intricacy of warehousing processes and the complexity of warehouse administration, which is due to the quantities of commodities processed, the type of technology used, and other warehousing operations. The paper by [10] presents the framework and classification of warehousing operations using the literature review as a reference framework model. It demonstrates the entire operations chain from product delivery to a warehouse to sorting, packing, storing, and dispatch.

The available literature research reveals that although various businesses, including warehouses of different types, have been analysed, the warehouse management process turned out to be the same. Warehouse management consists of primary warehousing measures—reception, capacity, order processing, and delivery.

The complexity of supply chain activities has obliged logistics enterprises to improve information flow both internally and externally. An integrated logistics information system is a unique business management model that monitors the flow of materials, information, and goods from the point of origin to the point of delivery, where all management actions are interconnected. In the works [11,12,13,14] the authors examine the possibilities of monitoring and tracking objects in warehouses using LPWAN technologies. The general idea is to ensure customer satisfaction by reducing the associated costs [15]. An integrated logistics system provides a combined solution for the management of warehouses, transport, and materials, integrating different functions of the business process including: (1) production function; (2) supply and storage functions; and (3) marketing function.

An integrated logistics information system requires a support system for implementation. Hence, the implementation may take several years. Such a support system is known as a logistics support system. Integrated logistics support (ILS) is an integrated and iterative process for adopting a support strategy that optimises operational support, leverages the existing elements, and pilots the system engineering process to quantify lifecycle costs and decrease the logistics footprint, making the system convenient for support. This process encourages companies to maintain or improve the standard quality of materials, to raise capital, to employ highly skilled staff, and to adopt the latest technology and information system to achieve the aims of their organisations [15].

Authors [16] in the paper discussed the basic principles of technologies, including their architecture, protocols and consensus algorithms, characteristics, and the challenges of integrating them. Furthermore, studies of the IoT device gateway, IoT information systems, management systems, cloud environment, and fog computing have been carried out (4.65% for each separate application). Authors identified significant gaps and future considerations that can be taken into account when integrating blockchain technology in the IoT domain.

Hence, the whole concept brings benefits and efficiency to companies. However, as a dominant element, support equipment increases the complexity of the process, as increasingly more devices and software packages are used. Costs associated with implementation, maintenance, skilled labour force, facilities, and vulnerability may be so high that some companies cannot afford to adopt this system.

IoT has numerous applications to offer to logistics and supply chain management and will have a huge impact on the future of logistics management. The objectives of IoT are to create a common network infrastructure to facilitate the ease of exchange of goods, services, and related information by machine-to-machine and human-to-machine interaction. IoT is the most interesting and essential area of research and is gaining attention worldwide.

For instance, a survey revealed that the use of computer-based packages for logistics and warehouse operations in Korea increased linearly in 1993–1995. These systems are commonly used for inventory management, material flow, sales, and demand forecasting [17]. However, electronic data interchange (EDI), which is a computer-to-computer data exchange of business documents in a standard electronic format between businesses, has been gaining popularity in recent years. EDI systems are used for order-related operations and transportation management. Yet, traditional ways of information exchange, such as fax, telephone, email, or personal visits, are also popular. In Korea, logistics and warehouse information systems are still in progress, and local IT service providers are struggling because of a lack of commercial interest. The author claims that medium-sized companies are unable to adopt modern IT systems because of financial issues [18]. In Korea, a new indoor positioning system (IPS) was developed by [19] and integrated with the warehouse management system. The system uses warehouse management tags to exchange data related to product availability, stock time, delivery time, and stock control. The author argued that this flexible system requires installing a few devices, such as a LED screen, a receiver, a transmitter, a gateway, and a WMS tag, and having integrated it with WMS, it will allow to easily monitor stock data at warehouses.

The paper published by [20] introduced an innovative warehouse management solution that includes the integration of artificial intelligence and the algorithm for storage planning, reception, and delivery of orders. This advanced solution requires barcoded pallets and shelves, and the use of GPS.

The work by [21] introduced an automated solution for warehouse monitoring based on IoT that includes ZigBee technology, sensors, situation handing system, and a wireless intelligent control system. This innovative system instantly controls heating, temperature, and humidity at a warehouse. Computers analyse data collected from sensors, and in case of an emergency, the system automatically calls the manager or emergency numbers.

A report by [22] from the United States analyses the role of IoT in supply chain management. The report states that IoT devices are mostly used to monitor inventory management, and RFID tags are widely used by supply chain companies. IoT devices are also used to monitor the temperature, humidity, and pressure of goods during shipping.

A study conducted in Malaysia [23] revealed that traditional SAP and WMS software packages integrated with IoT are used widely (Table 1). In warehouses, RF readers are used for material management, route planning, shipping operations, and dispatch. RF smart devices are used for operational functions under real-time radio communication.

The paper published by [25] describes the application and development of IoT in various sectors in China. IoT is used in inventory management, e-commerce logistics, distribution management, and order-related operations. Smart pallets and racks are used for optimising the storage space. For instance, in an inventory management solution based on industrial IoT and RFID, every single product in stock that is to be tracked receives an RFID tag. Each tag has a unique identification number (ID) that contains encoded digital data about goods, including their model, batch number, storage location, etc. RFID readers scan these tags. After scanning, a reader extracts tags’ IDs and transmits the related real-time information to the cloud for processing or display on mobile tablets.

IoT is an opportunity to solve complex tasks associated with the warehouse and logistics industry. In developing countries, companies are struggling with IoT adoption because of financial issues, while giant logistics and e-commerce companies are investing increasingly more into IoT development.

In logistics, the main idea was to develop a system to support all distribution activities to ensure efficiency and provision of real-time information in operations [26].

The use of information systems in the logistics sector dates back to the 17th century. Since then, it has gone through different improvements and advancements.

For instance, according to [27], the use of information systems in logistics has evolved through four stages. In the first stage (Industry 1.0) of the development process, logistics started using mechanisation of the transport service that includes using better transportation facilities in logistics. Later, technologies improved and changed following different phases of the industrial revolution. For instance, phase two (Industry 2.0) of the industrial revolution brought cargo automation and process coordination, while phase three (Industry 3.0) focused on industrial robots and information automation for the digital world. The term Industry 4.0 was first introduced in 2011 in Germany. The phase four (Industry 4.0) revolution involves the integration of ICT in production processes and activities. Smart factories have automated processes, digital operations, and information support on all organisational levels. The new scope essentially considered products to be highly trackable due to the transparency process of all involved parties, starting from dispatch to the end-user [28].

In the existing literature, different studies proposed different types of application of information technologies that can be used to improve the logistics sector in general and warehouse management in particular. The conference paper by [29] explained modern information technology and its applications in logistics and warehouse management. There are five major IoT technologies: RFID, wireless sensor networks (WSN), middleware, cloud computing, and IoT application software. IoT technologies have also been globally used in numerous industries, including warehousing management, logistics, and supply chain efficiency by providing information that is more detailed and up-to-date.

IoT gadgets include sensors and programmed ID hardware, such as RFID and barcode readers. Refs. [30,31] say that warehouse management involves huge quantities of products, staff, and operations, while managing it manually significantly encumbers the process. In this context, articles suggest that IoT integration will bring a revolutionary solution for monitoring and managing various warehouse operations through a central IoT warehouse management system. For example, installing sensors on shelves can help to manage the flow of goods, predicting real-time inventory. Additionally, Satti et al. [32] added that the practical application of IoT in self-driven forklifts in warehouses and IoT based warehouse robots minimise human efforts. Moreover, it allows tracking vehicles by creating a network of connections. The article by [10] states that modern technology should be implemented for material handling, order processing, and other warehouse operations. The article proposed a model for modern warehousing to solve the complexities of warehouse processes presenting the operational and tactical level problem.

IoT can be used for management and optimisation of end-to-end user activities, for instance: (1) Real-time visibility of inventory; (2) Remote monitoring of goods; (3) Compartments, pallets, or delivery status; (4) Shipping schedule; (5) Reception and storage operations; (6) Staff intensive activities; (7) Warehouse security and authenticity.

The findings of the literature that was studied recommend that a customised software package would be the best solution for warehouses with a significant number of order lines processed every day and high stock levels.

New innovative ideas that can boost the efficiency and functioning of a modern warehouse are studied eagerly, because industries opt for IoT solutions [33]. The principal idea of IoT and its strategic role in altering the approach towards warehouse management externally and within a company has been widely discussed in literature. IoT and its application are becoming increasingly more important fields of research.

According to [34], a warehouse with the RFID tag IoT technology demonstrated time saving of 81–99% in joint ordering, 100% time saving in processing, and real-time accurate inventory count. Other recent literature argues that IoT helps to maintain proper and efficient functioning in warehouse management. The article [35] describes the principle of IoT and its implementation in a warehouse for tracking, sorting, and monitoring. According to this article, IoT shows promising results in inventory and real-time management of the entire system. Implementing IoT in warehouses brings significant benefits and removes manual work. Ref. [36] revealed that the analysis of the data collected from IoT devices allows for a more accurate forecasting of demand, helps to solve security and authentication problems, and makes warehouse management more agile and visible. The article [37] proposed a warehouse management system integrated with fuzzy logic technology for key operations. The authors write about RFID tags on goods, pallets, and pending deliveries, presenting how such warehouse functions as order processing, inventory management, warehouse productivity, and system evaluation are handled with promising results. To synchronise and optimise inventory, data and information are entered in the intelligent inventory management engine to handle order change and processing problems. This includes data clustering and some machine learning methods, as well as the fuzzy inference system used for information processing in decision support. The output is then transmitted back to the host application and shares the results with mobile apps. Hence, the staff working with this IoT-based warehouse management system can receive the corresponding action information [37].

The IoT-based warehouse management system is a good approach for inventory management [38]. WMS gives promising results while IoT is integrated with the flow of goods at a warehouse and is used for stock monitoring, product location, tracking, and monitoring stock in real-time operations. The study [19] suggests a methodical warehouse system based on IoT, arguing that WMS should be integrated with IPS using Bluetooth and Wi-Fi. The recommendation is to develop a system using WMS tags and a software package to exchange data on commodities at the warehouse, which will control the location of goods, inventory, delivery time, stock time, and data on the surrounding environment. A few devices are needed to install tags, receivers, transmitters, and a gateway for a reliable warehouse management system.

The article by [20] introduced an innovative warehouse management solution that includes the integration of artificial intelligence and the algorithm for storage planning, order reception, and delivery. This advanced solution contains barcoded pallets and shelves, and shipments using GPS. In their work [21], authors developed an automated solution for warehouse monitoring based on IoT that includes ZigBee technology, sensors, situation handing system, and a wireless intelligent control system. This innovative system instantly controls heating, temperature, and humidity in the warehouse. A computer analyses the data collected from the sensors, and in case of an emergency, the system automatically calls the manager or emergency numbers. The article [39] demonstrates the design and architecture of an advanced warehouse with a case study. The proposed solution can be used in an existing warehouse or in designing a new smart warehouse.

According to the literature review presented in this section, the research was aimed at the use of IoT and its benefits in the researched field, focusing on certain problems within a smart warehouse. Table 2 illustrates the concept of IoT according to the available literature.

Several studies reveal that IoT plays a key role in warehouse management, and it has the capabilities to improve the overall efficiency of a warehouse. However, theoretically, it has entirely positive effects but practically, it may have some negative impacts as well; thus the conclusions made were of two types:**Positive aspects of IoT integration in warehouse management.** It is believed that IoT in warehousing brings a number of benefits to companies. The adoption of IoT enables smart warehousing, brings revolutionary changes, and that is why big industries such as Alibaba, DHL, Amazon, or Bluedart have already implemented it in their inventory, logistics, and warehouse management. The paper [7] addressed some reflections relating to crucial aspects of logistics 4.0. The article supports the implementation of an intelligent system and database to share information through the IoT platform to achieve a modern automated system. This interconnected mesh system will interconnect all warehouse operations and the workforce. It will make warehouses proficient and more transparent. The thesis presented by [38] is a remarkable work that reveals an advanced system based on IoT for tracking and inventory. The author argues that the proposed system is able to track the location of all commodities and provide location-based information. This system can analyse the conditions of goods and real-time shipment shifting. Moreover, this IoT-based WMS has the capability to inspect stock, eliminate error rate at a workplace, and ensure real-time inventory. It makes warehouse management costs efficient, timely, and proficient in all operations. The article by [19] argues that ICT has become a new smart trend that brings a revolution into warehouse management. The author recommends an indoor positioning system using Bluetooth for monitoring orders. The author developed a system that uses: (1) Bluetooth tags to transfer the related product information; (2) A receiver that tracks the location of a moving object and location of pallets; and (3) The integrated software package and database sever that monitors the warehouse and exchanges data. The developed system can make the stock time, delivery time, tracking, and material management much more efficient. Implemented in a warehouse, it brings benefits. A case study-based article to analyse business performance [23] claimed that the IoT-based warehouse management system helps to maintain customer satisfaction, make product delivery efficient, ensure accurate visible inventory, and maintain a productive labour force. It was observed to make a positive impact on various warehouse operations by reducing operational costs. The research proposal by [49] indicates the blending of lean production and RFID technology to improve the efficiency of warehouse management. In this research, 10 million parts belonging to 10,000 different distribution centres were included. It was noticed that for joint ordering in warehouse delivery operation, 80–99% of time was saved having implemented RFID smart tags, and the total operation time can be further boosted by up to 91% with cross-docking.**Negative aspects of IoT integration in warehouse management.** Previous discussions revealed that IoT-enabled warehouses are more efficient, faster, and accurately operated. However, this technology also has some downsides. According to previous research, this advanced technology has drawbacks listed in Table 3.

The conducted literature review revealed that the majority of sources of literature presented a theoretical model and only some of them offered a technical explanatory model for research findings. Hamdy et al. [6] use the famous Cobb–Douglas regression method to estimate the productivity of a company by using some indicators. The overall aim is to examine whether IoT can help to increase the productivity of a company. To investigate the impact of IoT implementation, this research used a quantitative analysis method with secondary data. The Cobb–Douglas method was used to examine the change in a company’s output (value-added) or productivity as a result of a change in input (capital, numbers of workers, cost of works) [6]. The findings claim that IoT has a positive effect on productivity. Moreover, IoT integrations show a better performance in terms of labour productivity, return on assets, inventory turnover, return on equity, asset utilisation, and overall efficiency.

The paper by [40] used a qualitative research model. The objective of the research was to conduct a multiple case study on how IoT technologies can be used efficiently in warehouse operations. The multiple case research included six semi-structured interviews. Secondary data, such as company reports and website content, were collected. The interviews were analysed using a content analysis method. The study examined different companies to discover whether all the cases demonstrated the same findings. This case research revealed similarities and differences between the companies interviewed, also illustrating advantages and disadvantages of IoT use in warehouses. A cross-case analysis was conducted to find the usual results and analyse possible differences between the cases, recording the outcomes in the data analysis.

The work [54] examines the opportunities of logistics 4.0 implementation in the warehousing industry in Sri Lanka. This investigation was conducted using comprehensive and systematic methods to review the existing literature on smart warehousing, smart logistics, intralogistics, and Sri Lankan Third-Party Logistics (3PL) industry domains. Through effective categorisation and integrative analysis, the aim of the paper is to identify the ways of warehousing process improvements using the logistics 4.0 technologies while attaining operational excellence. The first step of the research was to find scientific literature relevant to the study. A web search allowed access to various research papers in the mentioned study area from different academic databases, such as IEEE Xplore, Science Direct, research gate, etc. At first, fifty articles were selected based on their title, abstract, and keywords. Each article was examined in depth to eliminate information irrelevant to the study, choosing forty-two articles for further review. Scientists discussed the knowledge gap remaining after comparing the articles, considering suitability for the Sri Lankan 3PL company. In the end, thirty-five articles were selected for analysis. To discover the actual gap, an in-depth analysis of the articles was conducted. To discover the actual gap of knowledge, a comprehensive literature review was conducted. Articles were selected according to the year of their publication, in the range from 2000 to 2018. All the articles were analysed aiming to find such measures as smart logistics/logistics 4.0 definitions, logistics 4.0 technologies and applications, smart warehouse performance in the context of logistic 4.0, and challenges and opportunities of third-party logistics industry in Sri Lanka.

Authors in the manuscript [55] explored in more detail two of the aspects linked to human factors, identified as key factors for IoT implementation by the maturity models in the literature: user attitude toward IoT and qualification of the technical implementation team. The work has focused on the opinion of involved stakeholders in each case. The survey explored the specific details of the recommended qualifications for professionals working in teams where the implementation of IoT (and connected aspects of security and data management) are intensive, such as in projects. Authors have shown that projects could be representative of the case of general IoT implementations. According to the authors [56] considering the challenge of quality assessment in IoT systems, this work addressed the topic of the automated non-functional requirements evaluation. The objective is to enhance the Testability non-functional requirement of IoT systems under development.

Human efforts have been made to improve warehousing for ages. The conducted literature review revealed that warehousing is not a simple operation for storing goods. It consists of various complex operations, handling and shipping of large orders, correct inventory, and docking, which make it a complicated process. Therefore, the findings of the literature studied recommend that warehouses processing large numbers of orders and extensive amounts of stock every day as well as other organisations engaging in warehousing operations should be best supported by customised software packages.

Understanding the potential which a IoT-based warehouse can bring to supply chain industries is important. The analysis of scientific literature has clarified that the concept of an IoT-based warehouse is not new. Both industrialists and researchers have put significant efforts toward developing a modern warehouse to obtain the maximum profit from this growing industry. The implementation of an IoT enabled warehouse is complex and multifaceted. IoT in warehouses must be implemented considering the economic, social, and environmental perspectives and the operational and tactical level problems. This includes the installation of various devices, human-to-machine, and machine-to-human communication in warehouse operations. Therefore, different measures for designing IoT-based warehouses must be studied.

The implementation of IoT in companies operating warehouses brings more positive aspects. Therefore, companies with IoT integration gain a competitive advantage in business, improve their financial indicators, increase customer satisfaction, and improve the quality of the services provided. Every warehouse company should introduce measures such as intelligent management systems, IoT-based warehouse strategies, and paperless inventories which contribute to reducing negative environmental impacts, eliminate error rate at the workplace, and ensure transparent functioning, an enjoyable workplace, and productivity. The complexity of warehouse operations and handling a huge amount of goods are important factors in deciding to introduce IoT tools in warehousing.

The essence of the problem being analysed is the fact that the use of IoT in warehouse companies is a potential and important one, but the security and data protection measures are not perceived and are not researched in the existing literature. Therefore, privacy measures and data protection must be considered and improved while integrating IoT in warehouses. The results of such a study should be relevant for warehouse strategies and action plans for strengthening common warehouse management.

In summary, it can be said that the empirical results give a good overview of IoT technologies used in different warehousing companies. IoT technologies can be implemented and can have a positive impact on warehouse operations. Nevertheless, disadvantages, such as financial aspects of data protection and security, need to be considered. IoT technologies can be used in all warehousing operations.

The extensive literature review presented above allowed the identification of the following main research gaps:

First, the existing research mainly documents the positive effect of IoT on companies′ performance in general and warehouse management in particular. However, none of these studies take into account the associated side-effects of using IoT. This means that even though using IoT might improve storage management and sales, the overall effect might not be positive. The overall effect of adopting IoT depends on costs and benefits that IoT could bring to a company. Conducting a cost–benefit analysis is critical in order to fully understand the effect of IoT on the overall performance of a company; however, no research was conducted (to the best knowledge of the researcher) that aimed at evaluating the general effect of IoT. Thus, research conducted so far can be considered to be a partial analysis only. For instance, the use IoT might require a skilled labour force, which increases companies’ expenses.

Secondly, previous studies also do not differentiate the effect of IoT on companies of different sizes. The effect of IoT on a company’s performance could be context specific. Thus, a failure to consider differences such as the company size suggesting a positive impact of IoT on companies might be misleading.

Therefore, in this research, the researcher will fill the aforementioned gaps by conducting a cost–benefit analysis of adopting IoT, taking into account the company size.

### 2.2. Methodology and Research Design

The original idea was to collect statistical data from 500 companies from the European Union region and to examine how costs and benefits of IoT affect warehouse management, but the companies showed no will to participate in the research. Lack of statistical data led to a qualitative empirical research approach. In empirical research, the outcome of the research derives from tactile empirical evidence. Empiricism is the best platform to obtain knowledge through direct observation and experience rather than logic or reason alone. The main advantage of empirical research is the fact that it sheds light on threads to validate the received results [57]. The main idea of such research is to investigate the views, trends, and experiences on a specific matter. Moreover, empirical research is used to validate or authenticate previous investigations, making a study more authentic and precise [58] because it is built on observations and experiences.

The empirical approach raises a hypothesis and then tests the hypothesis against collected data. The following steps explain the empirical cycle [57]:Observation: This phase involves observing premises and studying existing literature reviews to gain knowledge about what has been done in the particular field, also analysing if the existing literature is not limited in scope or if it can be replicated, and if it can be used to support the research or be used to create a questionnaire for current research.Induction: The inductive framework is used to support a general conclusion from the data gained through observation. This phase helps to raise a hypothesis or objective of the research.Deduction: This phase helps the research to draw a conclusion from the assumptions made and to develop a research model. It highlights the metrics, variables, or logic to reach an unbiased outcome, developing a research model consisting of data collection and ways to collect that data.Testing: This phase includes tests and supports the hypothesis. The data needs to be analysed and validated using the appropriate analysis method. Depending on the research type, the analysis can be either qualitative or quantitative.Evaluation: In this phase, the data collected in the research is presented, supporting conclusions, identifying limitations, and making recommendations for further research on the same topics, and replicating final results. Both qualitative and quantitative research approaches can be used in empirical research to collect data [59]. There are several methods to collect data in qualitative or quantitative research, such as: (1) Case study; (2) Observation method; (3) Expert interview; and (4) Focus group.

To obtain significant insights into the topic, the existing literature, including scientific journals, reports, and websites, was analysed. After gaining significant knowledge about the topic, a questionnaire for expert survey was prepared. For this research, semi-structured expert interviews were used to collect data, because they allow researchers to obtain precise relevant evidence if the right questions are asked. It is a conversational method for collecting authentic evidence depending on where the conversation leads [57]. According to [60], a semi-structured interview of experts in the field of interest can be conducted to collect the appropriate data. A semi-structured interview is also based on an interview plan that includes specific questions and a thoughtful sequence of them, also allowing the researcher to ask additional questions not included in the plan if he believes this could enrich the research [60]. The questionnaire contains both open-ended and closed-ended questions. The questionnaire is designed to collect data that would allow examination of and distinguish the cost and benefit metrics, also revealing how these variables affect warehouse management. The outcome of this research will show the impact of IoT on warehouse management in companies of different size.

In order to evaluate the objectives of the research, its participants must be directly involved in warehouse activities, hold a managerial position, and have a higher education. Based on these criteria, nine experts were chosen to participate in the survey, as 5 to 10 participants are mandatory to ensure the reliability of survey results. A higher number of experts does not significantly affect the results [61]. The experts selected for this research have extensive experience in academic and industrial fields, and in IoT application development, warehouse, and logistics management in particular.

In order to evaluate the proposed conceptual model for the adoption of IoT in warehouse companies, a survey method was used. A detailed questionnaire was sent to experts. Expert judgment is important, so assessing that the person interviewed had the appropriate experience, knowledge, and understanding was important. In this case, experts had to have good knowledge of: IoT; Application of IoT in warehouses.

Therefore, respondents were selected based on the following requirements:Master’s degree in management at the least.Deep knowledge of IoT and its application area.Minimum of 5 years of experience in the field.

Academic performance and experience were also assessed. Four respondents participated in the survey. Each expert was contacted personally and introduced to the progress and objectives of the study. Questionnaires were sent by email. A total of 6 questions were asked, 2 of which were closed-ended, and the other 4 questions were open-ended questions. Experts were asked open-ended questions to obtain their opinions, comments, remarks, or insights. They were also asked to evaluate whether the order of the elements in the proposed model is correct and to what extent the experts’ opinions are aligned in this regard. First of all, the ranking of the obtained results will be performed (Table 4).

For decisions to be taken, the experts’ opinions must be balanced. In the case of two or more experts, the consistency of their opinions may be determined by applying a concordance factor. In the calculation of the Kendall’s concordance coefficient (W), the experts’ assessments are ranked (Table 5). The experts’ opinions are considered to be concordant when W→1, and discordant when W→0.

It is difficult to achieve objective measurement when using expert judgement. However, objectivity of assessment is achieved by comparing it with a value that is considered fair and reasonable. By calculating the group average of the assessment, a benchmark value or function is achieved. For a decision to be made, the judgement of the experts must be consistent and not contradictory, so that experts with strongly diverging opinions are excluded from the group. This leaves the expert group with only qualified specialists who understand the problem at hand. In some cases, the opinions of experts are divided into more than one group. As the study aims at a single opinion, if two or more groups of opinions are formed, it is considered that a common expert opinion has not been reached. The concordance coefficient helps to check the consistency of the experts’ opinions. Mr. Kendall has proposed a method to identify assessments that are not in agreement. Several main stages can be identified, from which a scheme of concordance for each of the evaluation criteria is drawn up:A measure of the consistency of the opinions expressed in the study is selected.An exemplary model of contradictory opinions is developed.Calculate the distribution of the selected model measure.

Each of the experts received a questionnaire containing the criteria. The highest criteria score was equal to the total number of criteria. The experts were asked to give each of the criteria a weighting from 1 to the highest. The most important criterion received the highest score, and the least important criterion received a score of 1. Each criterion had to be given a score and could not be 0. Criterion m, scored by experts n. After performing these calculations, the importance of the criteria will be determined (Table 6).

A quantitative descriptive data analysis was carried out to formulate the results. According to [59], a qualitative analysis plays a crucial role in research. It is used to analyse the main characteristics and data variables by explaining them in a more illustrative form. Such a practice gives wide macro and micro views of data, illustrating how metrics interact and affect each other. The data set can be summarised, described, and presented in a graphical presentation, also as a frequency distribution and median, assessing their relationship within variables.

## 3. Results and Discussion

### 3.1. Analysis of the Research Results on the Impacts of IoT on Warehouse Management

This survey aims to examine the potential cost and benefits of IoT to warehouse companies. To achieve this objective, an expert survey was conducted. This section presents and discusses the results of the expert survey. As described in the earlier section of the research, the study used the semi-structured expert survey method for data collection. During the interviews, additional questions were also asked to discover the solutions to tackle the cost problem in the survey. Therefore, this result analysis section also describes the experts’ recommendations to address the costs. Based on this analysis, the conceptual model will be presented in the next section.

IoT and its benefits for warehouse companies: To measure the potential benefits of using IoT in warehouse companies, the research used possible key benefit indicators, including the order lead time, inventory accuracy, workforce productivity, and return on investment. However, after conducting the survey, it was observed that a few cost indicators had a positive impact.

The survey found that IoT will increase a company’s sales. When asked about the impact of IoT on the sales of a company, all participants stated that sales will increase. Companies can use IoT to assess the needs of end users and to customise the goods accordingly, also allowing for more transparency. Accord to the experts, digitalisation allows both customers and companies to have real-time and more frequent information on shipments. Moreover, conditions of storage of goods, such as humidity, temperature, and location, can be monitored remotely, ensuring that the goods are shipped under better conditions. Such a practice can generate additional revenue by maintaining a healthy relationship with current and future opportunities. A similar situation was described by [44]. The size difference of warehouse companies does not change the above analysis results as illustrated in Figure 1.

In terms of customer satisfaction, the research found little variation in experts’ answers as illustrated in Figure 1. For instance, 90% said IoT would increase customer satisfaction and 10% believe that it would not have any impact [17]. One respondent said that sensors and feedback reporting mechanisms help to maintain future interactions and good relationships with customers. A similar approach was raised by [23].

According to the survey results, one of the main reasons for digitisation is to achieve real-time visibility of operations of a company and establish a centralised platform. An amount of 95% of participants indicated that IoT would improve the accuracy of inventory as shown in Figure 2.

A better, faster, accurate flow of Information on inventory is the current assets of a company [17], which has a huge impact on a company’s efficiency. The centralised system allows the accessibility of all data, such as information on stock, from different locations with real-time visibility in case of multiple locations of warehouses, and IoT forecasts of sales allow to do this easily. However, 5% of participants believe that a manual physical count is necessary. They believe physical touch is necessary for warehouses. Similar scenarios were explained by [23,45,47]. One respondent said that, as a small-scale warehouse company, they mostly rely on traditional aspects. However, they have started implementing IoT technologies and have had positive results.

When it comes to the order lead time indicator, the survey found a small difference of experts’ responses as illustrated in Figure 3.

For instance, 100% said that the order lead time for small and medium-sized companies would be shorter, but 10% believed IoT would have no impact on order lead time for larger companies, while 90% agreed that it would be shorter. The proof of delivery (PoD), the trailer tracking system, and RFID render magnificent results. The trailer tracking system gives the real-time location of shipments. The survey responses indicate one good example of reducing accidents and the improved safety of a smart forklift. A traditional forklift raises the functional issue, especially when a driver has to find a particular item at a specific time in a warehouse. Such a practice reduces productivity [17]. The PoD speeds up information and documentation flow with real-time availability. The costs related to printing, shipping, fuel costs, and human error were reduced promptly. The master thesis by [40] explains resembling benefits with IoT adoption.

When it comes to maintenance costs, 90% of participants said that the annual maintenance costs related to IoT systems would be low for companies of all sizes (Figure 2). However, 10% believed that they would increase for SMEs. A similar percentage of respondents said that there would be no impact on large companies. The IT experts working with IoT applications said that maintenance costs varied from 1 to 5% of the total application cost. Such maintenance costs include updates, patches, licenses, and certificates of IoT gadgets which can easily be adopted by warehouse companies while adopting IoT platforms. Participants stated that IoT had significant positive impacts on warehouse operations and on cutting extra costs.

The surveyed IT experts said that IoT-enabled predictive maintenance can reduce costs of overall warehouse operations. Sensors are installed in a smart pattern, which can detect a problem before it occurs or affect the process or downtime. Savings associated with IoT come in several ways, including:Improved productivity, which reduces workforce and expenses.Power and heating monitoring in warehouses, remotely monitoring energy expenditure and time savings.Total asset costs can be reduced with reduced work of gadgets and maintenance.Optimisation over time and speeding up the management of information and material flows reduce costs over time.Precise control of inventory.Energy savings.

A similar situation was explained by [41,44]. To achieve such saving goals, experts suggested adopting digitalisation completely.

All participants believed that the workforce feels comfortable and engaged in an IoT work environment. More work done in less time and extra time make the workforce innovative. Hence, it leads to reduced costs and higher efficiency at the workplace. The IoT work environment helps to retain workforce and increase morale in both small and large enterprises. However, Figure 3 reveals that 20% of experts from the survey said that IoT has some social impacts. They believed that technology was replacing people and stealing their jobs. The remaining participants showed a mixed response.

They believed that technology was stealing jobs but increasing revenue, believing that a rich economy would offer more jobs in the future. The article by [6] also claims that IoT implementation has a positive impact on the financial performance of companies, and their market value, profitability, and labour productivity.

When it comes to IoT and its costs for warehouse companies, participants believe that there are substantial intangible costs associated with IoT other than costs of installation. For example, there is always a shortage of a skilled workforce in IoT and compatibility of the IoT eco-system. Figure 4 shows that 90% said that there will be a shortage of a skilled workforce in companies of all sizes, and 10% believed there was an available workforce on the market.

All participants said that the privacy of confidential data and customer information must be a top priority in IoT.

As per Figure 4, in terms of data privacy and cyber-attacks, a total of 90% of participants identified the main obstacle to IoT implementation in larger companies, which is increased costs related to data security breaches and data management. An amount of 77% said that such costs would increase investments for SMEs. Barreto et al. [7] made a similar observation. Companies need to earn customer trust regarding the confidentiality of their data. These costs increase proportionally to the size of the company. In contrast, the surveyed IT experts (22%) said that for SMEs, a variety of secure encrypted platforms can be used to tackle such costs, but all participants believed that these costs would be high because larger firms are more prone to cyber-attacks.

Like any other investment, IoT-related investment requires planning and careful consideration for implementation. Small-sized enterprises must have a planned strategy on the part of the operations to be digitalised using IoT. Figure 4 shows that 77% of participants said that the IoT platform has high installation costs for both SMEs and LEs, which increase with the size of the company. However, 30% of participants stated that installation costs were a one-time investment. Growth comes at a price. In contrast, the surveyed IT experts (22%) said that there were affordable IoT gadgets on the market for SMEs. As stated before, IoT implementation requires proper planning. Companies should have a clear vision and strategy when adopting advanced technology. They should analyse the adoption strategy by finding out which section of the warehouse needs to be integrated with IoT. Experts suggest for SMEs to adopt IoT in several steps as some costs may vary and cannot be foreseen.

All participants believe that IoT projects have a good return on investment, allowing them to feel benefits faster.

However, as per Figure 5, 77% of participants said that for SMEs, RoI on IoT platforms is a capital investment and has a longer payback time, but all participants believed that a return on investment for IoT platforms can be achieved in larger companies. Regardless of the company size, benefits from innovation are significant. According to the survey, RoI for companies of all sizes will manifest through:Increased efficiency in every management operation.Accurate inventory.Energy savings.Ready-to-access data available for forecasting.

The paper by [44] explained that the adoption of IoT systems increases gross margin, inventory turnover, market share, return on sales, and reduces sales, general, and administrative expenses.

To sum up the results of the study, apart from a few obstacles, IoT can be said to have a positive impact on warehouse management for companies of all sizes. The presented literature review gives a wide overview of IoT technologies which are used in different warehousing companies. The survey found numerous benefits of the use of IoT in warehouse companies. To sum up, IoT brings more revenue along with making warehouse functioning convenient and productive. Responses to the survey reveal that IoT has a huge impact on customer satisfaction and company sales. Customers are considered the backbone of every company. Regardless of the company size, customer satisfaction brings significant revenue to companies. A similar scenario was also confirmed by [65] in the literature review section. The survey revealed that in the case of SMEs, the payback period may be longer, but larger companies can feel the benefits sooner. Moreover, when asked about the impact of maintenance costs related to IoT, participants agreed that they would be lower, also cutting down other management costs. The benefits examined in the data analysis section outweigh the costs, also bringing additional intangible benefits.

The research refuted the hypothesis that IoT has different impacts on warehouse management depending on the company size.

However, when it comes to the second hypothesis that IoT would have different costs and benefits in companies of different sizes, the research partially authenticated it, because the experts believed that the costs would vary with the company size, but the benefits would be similar.

When making a decision to invest in IoT, companies have to consider many factors that play a major role. These include installation costs, data security, and shortage of a skilled labour force. Companies have different scenarios to consider when going digitalised, because IoT is capital intensive. A similar situation was explained by [53]. However, some participants believe there are several possibilities to reduce these costs, for example, by using a third-party data cloud service or versatile encrypted security to secure the system. Yet, data protection and its financial aspects will remain the main concern for companies, as there will always be weak loop points for cyber-attacks in IoT systems. For larger enterprises, such costs will be higher. Meanwhile, 80% of the participants agreed that SME data is more secure, as hackers mostly target larger companies and larger companies are the ones that entrust data to third-party service providers.

Taking into account the results of the conducted research, and in order to assess how important the interaction of these criteria is in the development of the model, we evaluated their dependence (Figure 6).

The obtained results show that “Customer satisfaction” and “Return on investment related to IoT systems for SMEs and Les” have the greatest interdependence.

The presented literature review and the results of the survey allow the conclusion that IoT brings more benefits than costs regardless of the size of the company. Theoretically, IoT has tremendous potential, but its installation costs and privacy issues have to be tested practically.

### 3.2. Relevance of the Model and Tools for Its Development

Traditional warehousing was not designed to handle large amounts of customer orders for immediate delivery. Manufacturers, retailers, warehouses, and logistics providers are all struggling to adjust to meet these needs, which has led to more extensive use of modern innovation to tackle these difficulties.

Having analysed the impact of IoT on warehouse management, this research also aims to contribute to the development of a conceptual model for IoT implementation. The model offered in this section of the research is based on the survey results. Having raised the hypothesis, it was observed that the main obstacles identified during the research were installation costs, data privacy issues, and a shortage of a skilled workforce. The data privacy issue will remain the main area of concern now and in the future. Therefore, the main idea behind this model is to provide solutions for issues examined in the results section and to propose solutions recommended by the experts. The benefits of IoT examined in this research substantiate the implementation of IoT in warehouses. The steps to be taken to implement IoT consider IoT installation costs. This model is proposed based on the issues raised during the survey and the recommendations made by the experts [17].

Additionally, after getting the insights of IoT utilization during the literature review section of this research, the utilization of IoT in different warehouse activities was also added. This contributes to how and where IoT can be implemented in warehouse operations to improve functioning (Figure 7).

**Adoption activities**. The aim of this section is to show where and how new technologies can be integrated in order to improve the process and thus provide real benefits to users of these services [17]. The warehouse functions mainly include the following:


**1. Warehouse functions include:**
**Reception**: This process involves the reception of commodities. To function properly, it is necessary to check if the correct quantity of goods was received at the right time. To this end, RFID tags are highly accurate and reduce human errors. They document the arrival of goods. RFID integrated with an automated scanner can capture the weight and dimensions of a package, indicating a proper location for its storage. This improves the functioning of the reception process, allows to unload the dock quickly, and to clear the space for other shipments.**Put-away**. This is the process of the movement of goods from the reception dock to the most optimal storage location. Mobile and wearable devices take the inbound goods to the right location to incorporate inventory coherently. IoT enables smart forklifts that can reduce accidents during this process, thus reducing the time needed to complete tasks and optimising storage.**Storage**: Storage in a warehouse means putting commodities in the most appropriate location. Warehouse space is capital intensive, and space optimisation can reduce this cost. IoT enables HCL warehouse space optimisation solutions, offering real-time location-specific optimisation and reducing the space turnaround ratio. RFID tags also store the location-related data of the product.**Assembly and shipping**. This involves processing customer orders and shipping. This process requires high efficiency, as errors reduce customer satisfaction. Wireless wearable devices can make such a process optimal, allowing for real-time scanning within the entire warehouse. Smart forklifts with sensors and scanners transmit data related to precautions to handle products along with outbound and inbound deliveries. GPS enables the trailer tracker system to indicate the real-time location of the shipment. The PoD solution can be used for real-time delivery reports.


**2. The warehouse process includes**: inventory, monitoring, logistics, forecasting, and risk mitigation. The IoT technology for monitoring allows for predictive maintenance. Environmental sensors transfer data to ensure proper storage conditions of goods. The IoT integrated warehouse management platform can enable a sustained flow of real-time information on inventory levels, trucks, cargo, drivers, and location. It could increase inventory accuracy and mitigate risks, such as a fire or a theft. Solutions such as RFID, wearables, and sensors provide information on real-time inventory, location of goods, and data for future sales and inventory forecasting.

Level of IoT adoption. The experts surveyed agreed that installation costs are the main obstacle to digitalisation in companies of all sizes. Experts recommended adopting IoT in a few steps rather than at once in warehouse operations. This would give time for a company to analyse its potential benefits to the company. Early adopters are recommended to identify the warehouse operations where IoT adoption is necessary. The following steps are recommended in this model:Step 1: A company should constantly monitor customers’ expectations. The results of this research show positive impact of IoT on customer satisfaction. In step 1, such a technology could be used for monitoring, risk mitigation, e-commerce website, and mobile application for the sale of products. Companies should use IoT for an extensive analysis of customer needs to initiate new products and services.Step 2: Once the potential benefits have been identified, the company firm should integrate the IoT platform in the existing warehouse activities; for example, integration of IoT with the warehouse management system, enterprise resource planning (ERP), SAP, PoD, and light robotics.Step 3: This step of adoption involves high-tech integration and the idea of a fully automated warehouse. Such a warehouse works with automated forklifts, drones, artificially intelligent, robots, and autonomous vehicles. There are well-functioning human-free advanced warehouses in China and the USA.

Shortage of skilled labour force. According to the surveyed experts, companies and the government should take initiatives to encourage and educate people through apprenticeship programs.

### 3.3. Results of the Expert Survey

Experts evaluated the model of application of green logistics measures in transport companies positively. They emphasised that digitalisation increases the company’s competitive advantage, productivity, operations, profits, and customer satisfaction. Experts believe that these elements play an important role in encouraging companies to use smart technology. However, one expert said that smart technology should be adopted as soon as possible to stay competitive in the market.

The experts agreed that installation costs, data privacy, and a shortage of a skilled workforce were the main obstacles for companies to go digitalised. The experts also pointed to the complexity of IoT systems with interconnections of several devices and compatibility of such systems with the existing frameworks.

Another important point is a government’s initiative for technological progress. Respondents mentioned that without such an initiative, the implementation of smart technology measures in warehouse companies becomes difficult, because IoT can be a capital-intensive investment, and small and medium-sized enterprises may have problems with funding. Government subsidies could be a possible solution.

Experts noted that a warehouse company cannot implement smart technology without human resources. There is a shortage of IT and technical experts in particular. People should be encouraged to acquire technical education and skills. Thus, the experts recommend introducing apprenticeship programs in companies. Moreover, the government should take initiatives with technical education programs. Another major factor identified during this survey is the maturity of the technology. IoT technology is relatively young, making it difficult to develop a coherent workflow for product development. Small companies do not have much experience and knowledge with IoT implementation. The participants also believe that it is easier for large companies to adopt IoT measures due to their higher financial capacity.

To sum up, the experts believe that the proposed model provides a good scenario for IoT implementation. Moreover, the experts identify that the following factors are also worth consideration:Time necessary to go digitalised.Maturity of technology.Compatibility of the system.

Considering the positive assessment of the model by experts, it is important to assess whether the elements of the proposed model are presented in sequence and their order is suitable. Therefore, after applying the expert evaluation method, the ranking of the obtained results and its calculation was performed (Table 7, Table 8, Table 9 and Table 10).

After ranking, concordance coefficients were calculated (Table 11).

The obtained results show that the experts’ opinions are aligned. Therefore, it was important to evaluate how the elements should be arranged in the constructed model according to their importance (Table 12, Table 13, Table 14 and Table 15).

The results of the expert evaluation influenced the corrections of the created model (Figure 8).

After the expert evaluation, the main changes in the order of elements in the model occurred in the first two blocks (1st block: Internal elements and External elements; and 2nd block: Warehouse functions and Warehouse processes).

Taking into account the results obtained during the study, the following benefits of the model can be distinguished:This model contributes to the analysis of the benefits and costs of IoT. Finding solutions to fund these costs would encourage companies to implement IoT.The proposed model describes the functioning of a warehouse, indicating how and where IoT technology could be implemented and identifying the impacts of IoT implementation on warehouse activities.This research also contributes to the levels of IoT adoption to tackle the installation costs. For example, a company could start with basic automation and having identified the potential impacts, it could move to the second and third steps.The presented model also describes the possible solutions for data privacy. For instance, a company could hire third-party IoT system security providers in the first step of adoption.

During the development of this model, the main focus was on the key performance indicators (KPI) of warehouse management and the capital-intensive factors, such as installation costs. Warehouse KPIs were described and selected in the literature review section of this research. Theoretical models were developed without considering the real environmental elements and economic conditions that affect cost savings and efficient business processes. Still, the proposed model determined that the implementation of IoT requires significant resources, including human (skilled labour force) and financial resources.

Thus, the presented literature review and the results allow for the division of the main elements that advocate for adopting IoT into internal and external ones. The internal key elements are inventory, order lead time, workforce productivity, and operational indicators. The external parameters include sales, customer satisfaction, competition, and supplier relationship. The results reveal that IoT shows a positive response to these parameters, and the surveyed experts believe that IoT should be adopted.

## 4. Conclusions

Warehouses have a significant role in modern society, playing an important role in logistics and the whole supply chain. For any organisation to stay on the market, it is necessary to ensure a smooth movement and operation of the entire chain. Inefficient organisational processes may lead to a loss of time and money. This may also be due to the inefficient use of human resources or insufficient automation of business processes. It should be noted that global trends indicate that the complexity of warehousing process will increase in the future. This means that the need for an efficient system and the customer-oriented approach will be predominant in the future.

The concept of the industrial revolution 4.0 is still young but is becoming increasingly popular. Researchers and individual industrialists have tried to explore it, as it can bring magnificent changes not only to industry, but also to society and the global economy. Warehousing is not a simple good storage operation. It consists of various complex processes, processing of huge orders, shipping, monitoring correct inventory, and docking. The presence of IoT in the warehouse means human-to-machine and machine-to-machine interactions. The role of human work and its planning in particular will transform entirely. The ability to correctly interpret and understand these changes will allow firms to gain a competitive advantage in the market and keep up with the latest technological trends.

The literature analysis shows that shifting from traditional warehousing to modern warehousing requires time. IoT integration allows companies to gain a competitive advantage in business, improve financial indicators, increase customer satisfaction, and improve the quality of the services provided. Every warehouse company must introduce intelligent management systems. The creation of an IoT-based warehouse strategy brings paperless inventory, which contributes to reducing negative environmental impacts, eliminating error rate at the workplace, transparent functioning, an enjoyable workplace, and productivity. The complexity of warehouse operations and the handling of extensive amounts of goods are important factors in determining the introduction of IoT tools in warehousing.

IoT has a lot to offer to logistics and warehouse companies. However, potential benefits and costs associated with IoT must first be examined. This research aimed to analyse if IoT has different impacts on warehouse management and different costs and benefits depending on the size of a company. Comprehensive results cannot be obtained without considering the size of a company. Such knowledge gaps cannot reveal the real impacts of IoT and its adoption. The outcome of this study revealed that IoT has positive impacts on the management of warehouses of all sizes. IoT implementation has a positive response on elements such as real-time inventory and its accuracy, predictive maintenance, operational efficiency, work quality, and productivity of a warehouse. Also, it brings additional benefits such as energy saving, improvement in sales, and customer satisfaction. The study also found that IoT has the flexibility to integrate into the functioning process of a warehouse. Companies then have a much more agile warehousing facility, ready to meet the preferences of new and existing customers.

The main costs of IoT include immense installation costs, costs associated with data security breach, data privacy, and data management. Such costs vary from company to company depending on their size and IoT implementation strategy.

The proposed model of adoption of IoT in warehouse companies is based on internal and external benefits, warehouse activities, and costs. Benefits always encourage the implementation of new technology. The proposed model describes the possibilities where and how IoT can be implemented along with the latest IoT platform for different warehouse operations. The potential benefits and costs were also identified based on different warehouse activities. When it comes to costs, the survey found that the slow implementation of IoT measures could be a possible option. Moreover, warehouse companies and the government should take the initiative to tackle obstacles such as a lack of a skilled workforce and data privacy.

Limitations of the research. There are two main factors that affect and limit the research: first is a limited time for collecting data. It requires a great deal of endurance to collect such data within a limited time horizon. The original idea was to collect data from 500 companies in the concerned field, later performing a multiple linear regression to examine how the selected cost and benefit metrics affect companies depending on their sizes. The second factor was a lack of statistical data, as selected companies showed no will to participate in the research. Therefore, the researcher ended up collecting data through an expert survey. The quality of such data could be biased since experts can have biased views, which can affect the final outcome. However, the participants were well informed that their individual responses would be considered as one comprehensive outcome. A semi-structured questionnaire was used to collect data. This way, the researcher could ask additional questions which were not in the plan in order to enrich the data collection.

Recommendations for future research. Having conducted a review of the existing literature and a survey, limitations of this study and recommendations for future research were identified. As mentioned before, the unwillingness of companies to participate in the research was the main limitation. Data collected from companies could lead to an entirely different outcome of the research. Specific users of the IoT platform would have more precise information on the impacts of IoT on warehouse companies and the costs and benefits for companies depending on the company size.

## Figures and Tables

**Figure 1 sensors-23-02213-f001:**
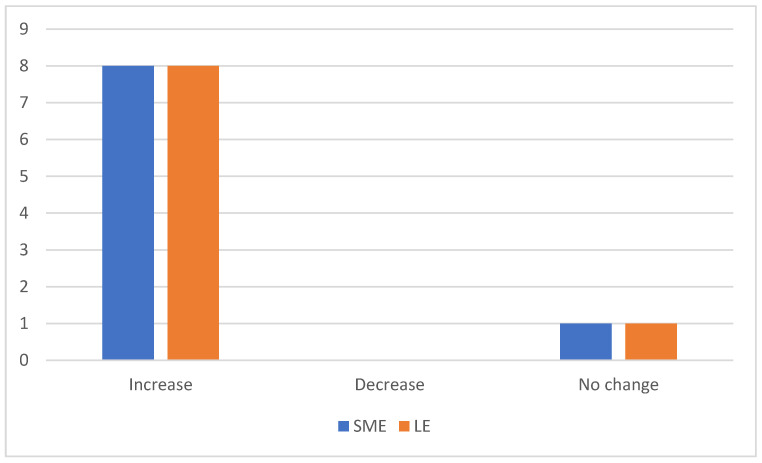
Participants’ responses to questions on the impact of IoT on sales and customer satisfaction in small and medium-sized (SME) and large enterprises (LE).

**Figure 2 sensors-23-02213-f002:**
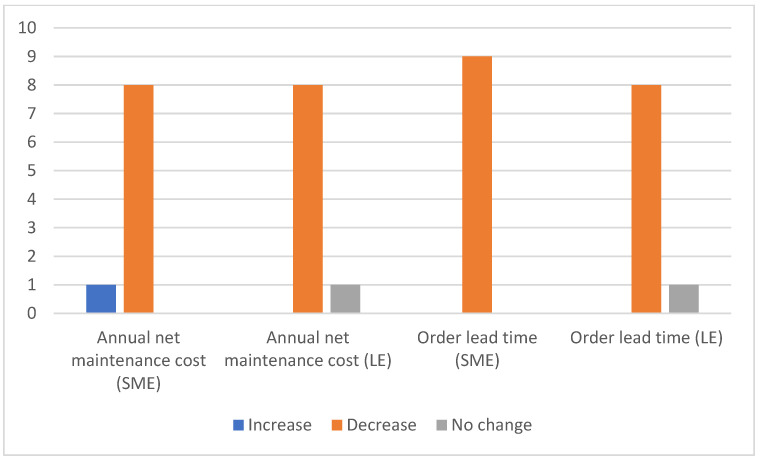
Participants’ responses on impact of IoT on inventory accuracy, annual net maintenance cost, and order lead time for SMEs and LEs.

**Figure 3 sensors-23-02213-f003:**
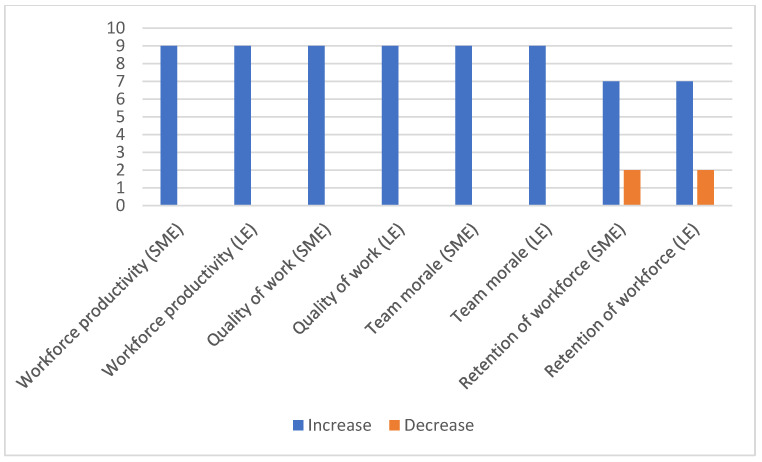
Participants’ responses on the impact of IoT on productivity, quality of work, team spirit, and retention of workforce for SMEs and LEs.

**Figure 4 sensors-23-02213-f004:**
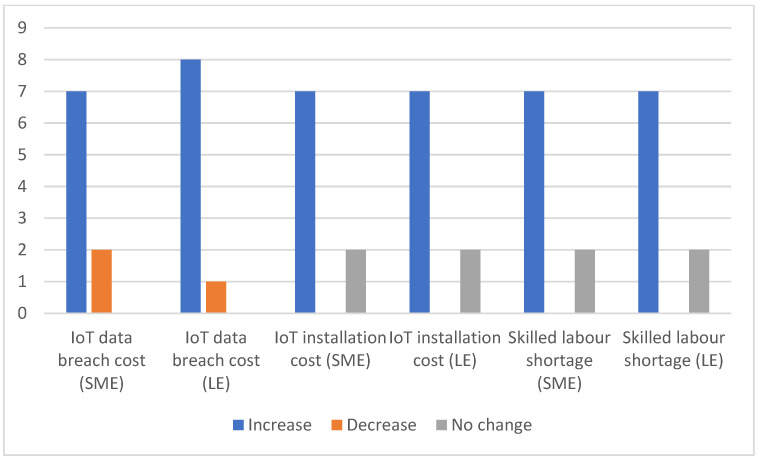
Participants’ responses on data breach costs, installation costs, and skilled labour shortage metrics related to IoT systems for SMEs and LEs.

**Figure 5 sensors-23-02213-f005:**
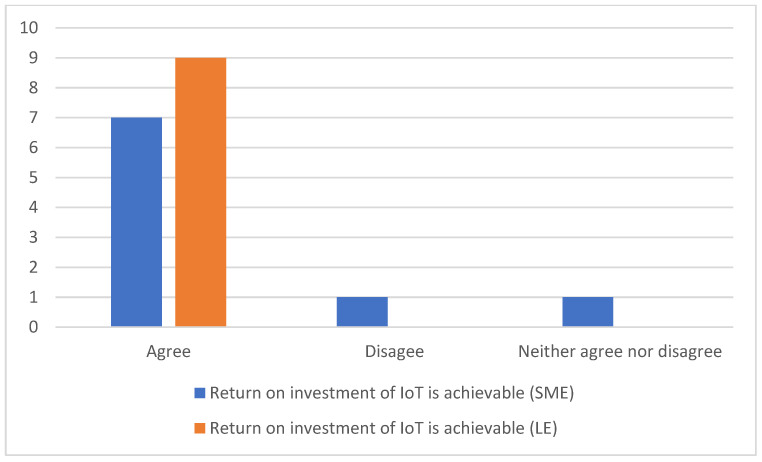
Participants’ response on return on investment related to IoT systems for SMEs and Les.

**Figure 6 sensors-23-02213-f006:**
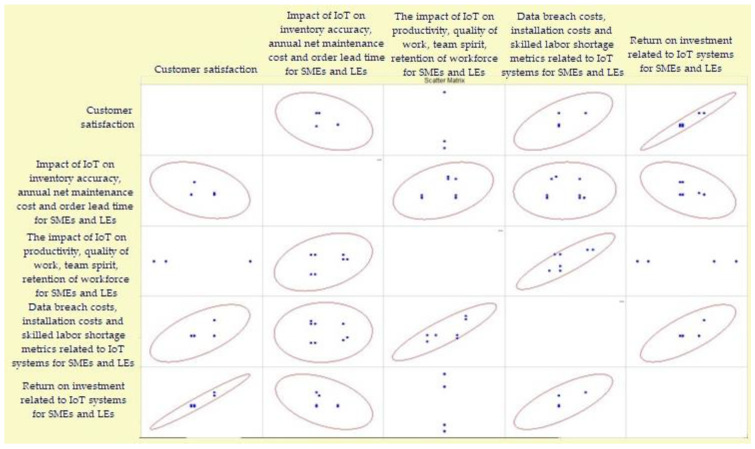
Interdependence of the studied factors.

**Figure 7 sensors-23-02213-f007:**
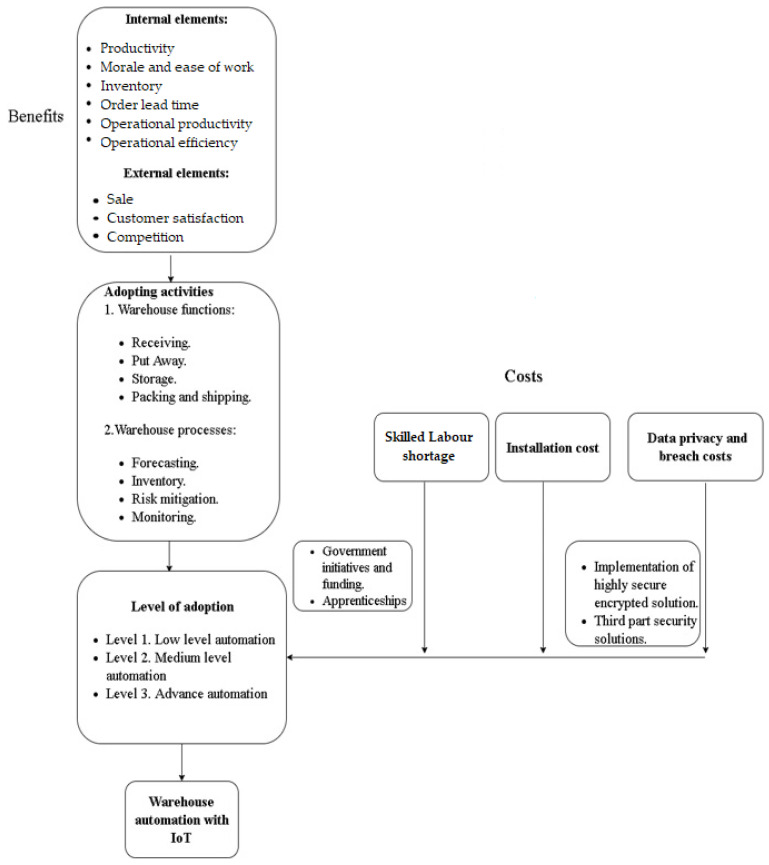
Conceptual model for the adoption of IoT measures in warehouse companies.

**Figure 8 sensors-23-02213-f008:**
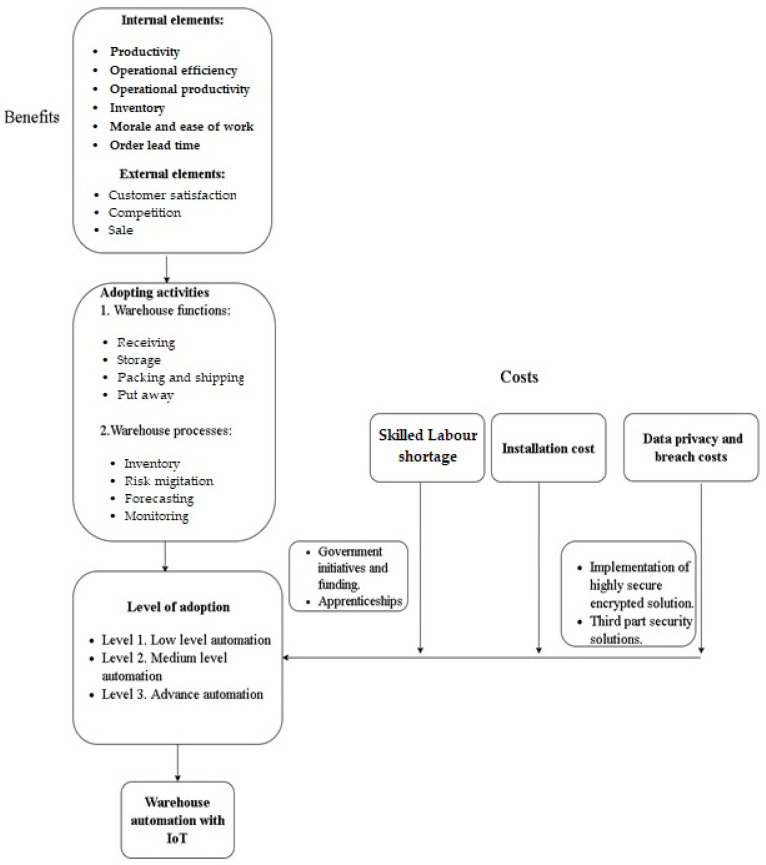
An improved conceptual model for applying Internet of Things tools in warehouse companies.

**Table 1 sensors-23-02213-t001:** Primary IoT technologies [23,24].

IoT Technologies			
		Inbound Applications	Outbound Applications
**Identification technologies**	RFID WSN QR code Barcode EDI Sensor GPS	Placement Cycle count Location of goods Receipt management Physical count Storage optimisation Workforce optimisation Cycle count Temperature, humidity control Material flow control Real-time inventory Sales and demand forecasting	Order acceptance from host systems Packing list Dispatch Prioritisation of orders. Transportation End-user services
**Communication technologies**	Zigbee Z wave MQTT Bluetooth Wi-Fi NFC WSN Middleware Cloud computing	Real-time information exchange Integration with warehouse management system packages.	–

**Table 2 sensors-23-02213-t002:** Concept of IoT used in scientific literature.

Problem Researched/Solved	Method Used	Findings	Sources
Integration of IoT in warehouse management.	Qualitative	This thesis filled the existing literature gap for the reception and shipping process at a warehouse.	[40]
Modelling a smart warehouse.	Domain analysis	Illustrated and validated the method and the proposed reference architecture.	[39]
A resource-based perspective on information technology capabilities and corporate performance.	Regression	A company with integrated IT tends to outperform on costs and profit-based performance measures.	[41]
Warehouse performance measure.	Content analyses	This article contributes to research development by proposing numerous future research directions on warehouse performance and evaluations.	[2]
Important aspects of IoT in supply chain management.	Bibliometric	Latest developments and trends in the use of IoT in various supply chain operations, highlighting several possible directions for future research.	[42]
Feature, semantic meaning, and descriptive model of IoT.	Descriptive	Two descriptive models about IoT were introduced.	[43]
Examining the change in manufacturing firms’ financial performance before and after adopting IoT in supply chain management.	General linear model	The adoption of IoT systems increased gross margin, inventory turnover, market share, return on sales, and reduced selling, general, and administrative expenses.	[44]
Impact of IoT on inventory and logistics management.	Explanatory	IoT is an emerging solution and brings efficient management of and real-inventory count.	[45]
Investigation and interpretation of business performance from multiple perspectives after the implementation of a modern warehouse management system.	Descriptive	Air shipping time decreased by 54%, inventory accuracy improved from 98% to 99.52%, and 11% less customer complaints.	[23]
Analysis of the performance of IoT implementation from the perspective of financial performance, productivity, and market value.	Ordinary least square regression	IoT implementation has a positive impact on financial performance, market value, profitability, and labour productivity of companies.	[6]
Classification of warehouse design and control problems at a strategic, tactical, and operational level.	Event research	The synthesis of models and techniques for warehouse design and development was pursued. Recommendations concerning the design-oriented approach were presented.	[10]
Researched effectiveness of designing warehouses to determine the average order processing time and the distance travelled using a data mining technique.	Simulation method	The proposed design with added tunnels in the warehouse improved order processing efficiency by 50%.	[46]
Review of literature on typical decision problems in design and control of manual order processing.	Analytical	Fewer research was conducted on the picker-to-part systems compared to part-to-picker systems. The number of publications on picking bays, batching, storage strategies, and sorting are minimal.	[47]
The issue of storing hazardous products in warehouses using IoT.	Event research	Product allocation planning with compatibility constraints (PAP/CC) was recommended, and it shows the improvement to unplanned product movement due to human error and reduces the size of floating location in a Tunisian chemical firm.	[48]

**Table 3 sensors-23-02213-t003:** Negative aspects of IoT integration in warehouse management.

Delivery Function	IoT Impacts	IoT Technology USED	Sources
All operations in WMS	CCTV footage steals customer and workforce privacy, security vulnerabilities, complexity of interconnection of thousands of devices, electronic waste.	Sensors, cameras, Wi-Fi, GPS, cloud-based network.	[22,50,51,52,53]
All operations in WMS	Network/system, personal data-stealing, hacking, or threads.	Sensors, cameras, Wi-Fi, GPS.	[7,22,50,52]
Online transactions	Third-party access threads, unauthentic transactions, high maintenance costs, unemployment, electronic hazard waste, environmental impacts.	Online transaction Software packages, cyber-physical system.	[7,22,53]

**Table 4 sensors-23-02213-t004:** Formulas for calculating ranks (source: compiled by authors based on) [62,63,64].

Formula	Name of Calculation
$\overset{n}{\underset{i=1}{\sum }}R_{ij}=R_{ij}$	Sum of ratings
${\bar{R}}_{j}=\frac{\sum_{i=1}^{n}R_{ij}}{n}$	Mean of ratings
$\overset{n}{\underset{i=1}{\sum }}R_{ij}-\frac{1}{2}n\left(m+1\right)$	Difference between the sum of ratings and the fixed value
${\left[\overset{n}{\underset{i=1}{\sum }}R_{ij}-\frac{1}{2}n\left(m+1\right)\right]}^{2}$	Square of the difference

**Table 5 sensors-23-02213-t005:** Formulas for calculating concordance coefficients (source: compiled by authors based on) [62,63,64].

Formula	Name of Calculation
$W=\frac{12S}{n^{2}\left(m^{3}-m\right)}$	The concordance coefficient
$\chi^{2}=\frac{12S}{nm\left(m+1\right)}$	A random size
$W_{min}=\frac{\chi_{\upsilon ,a}^{2}}{n\left(m+1\right)}$	The lowest value of the concordance coefficient

**Table 6 sensors-23-02213-t006:** Formulas for calculating the importance of criteria (source: compiled by authors based on) [62,63,64].

Formula	Name of Calculation
$\bar{q}=\frac{{\bar{R}}_{j}}{\sum_{j=1}^{m}R_{j}}$	The ratio of the average rank to the total sum of ranks
$d_{j}=1-{\bar{q}}_{j}=1-\frac{{\bar{R}}_{j}}{\sum_{j=1}^{m}R_{j}}$	Reverse size
$Q_{j}=\frac{d_{j}}{\sum_{j=1}^{m}d_{j}}=\frac{d_{j}}{m-1}$	Significance indicator
${\bar{Q}}_{j}=\frac{\sum_{i=1}^{n}B_{ij}}{\sum_{i=1}^{n}\sum_{j=1}^{m}B_{ij}}$	Criterion importance indicator

**Table 7 sensors-23-02213-t007:** The results of the sum of ranks of the expert evaluation.

Elements of Expert Evaluation		Criterion Encryption Model					
**Sum of ratings**	**Internal elements**	**Productivity**	**Morale and ease of work**	**Inventory**	**Order lead time**	**Operational productivity**	**Operational efficiency**
		9	19	13	22	11	10
	**External elements**	**Sale**	**Customer satisfaction**	**Competition**	**-**	**-**	**-**
		10	6	8	**-**	-	-
	**Adopting activities (Warehouse functions)**	**Receiving**	**Put away**	**Storage**	**Packing and shipping**	**-**	**-**
		7	11	9	10	-	-
	**Adopting activities (Warehouse processes)**	**Forecasting**	**Inventory**	**Risk migitation**	**Monitoring**	**-**	**-**
		12	6	8	14	-	-

**Table 8 sensors-23-02213-t008:** The results of the average of the ranks of the expert assessment.

Elements of Expert Evaluation		Criterion Encryption Model					
**Mean of ratings**	**Internal elements**	**Productivity**	**Morale and ease of work**	**Inventory**	**Order lead time**	**Operational productivity**	**Operational efficiency**
		2.25	4.75	3.250	5.500	2.75	2.500
	**External elements**	**Sale**	**Customer satisfaction**	**Competition**	**-**	**-**	**-**
		2.5	1.5	2.000	-	-	-
	**Adopting activities (Warehouse functions)**	**Receiving**	**Put away**	**Storage**	**Packing and shipping**	**-**	**-**
		1.75	2.75	2.250	2.500	-	-
	**Adopting activities (Warehouse processes)**	**Forecasting**	**Inventory**	**Risk migitation**	**Monitoring**	**-**	**-**
		3	1.5	2.000	3.500	-	-

**Table 9 sensors-23-02213-t009:** The results of the expert evaluation of difference between the sum of ratings and the fixed value.

Elements of Expert Evaluation		Criterion Encryption Model					
**Difference between the sum of ratings and the fixed value**	**Internal elements**	**Productivity**	**Morale and ease of work**	**Inventory**	**Order lead time**	**Operational productivity**	**Operational efficiency**
		−5	5	−1	8	−3	−4
	**External elements**	**Sale**	**Customer satisfaction**	**Competition**	**-**	**-**	**-**
		2	−2	0	-	-	-
	**Adopting activities (Warehouse functions)**	**Receiving**	**Put away**	**Storage**	**Packing and shipping**	**-**	**-**
		−3	1	−1	0	-	-
	**Adopting activities (Warehouse processes)**	**Forecasting**	**Inventory**	**Risk migitation**	**Monitoring**	**-**	**-**
		2	−4	−2	4	-	-

**Table 10 sensors-23-02213-t010:** Expert judgment difference squared results.

Elements of Expert Evaluation		Criterion Encryption Model					
**Square of the difference**	**Internal elements**	**Productivity**	**Morale and ease of work**	**Inventory**	**Order lead time**	**Operational productivity**	**Operational efficiency**
		25	25	1	64	9	16
	**External elements**	**Sale**	**Customer satisfaction**	**Competition**	**-**	**-**	**-**
		4	4	0	-	-	-
	**Adopting activities (Warehouse functions)**	**Receiving**	**Put away**	**Storage**	**Packing and shipping**	**-**	**-**
		9	1	1	0	-	-
	**Adopting activities (Warehouse processes)**	**Forecasting**	**Inventory**	**Risk migitation**	**Monitoring**	**-**	**-**
		4	16	4	16	-	-

**Table 11 sensors-23-02213-t011:** Concordance coefficient.

Concordance Coefficient Values	Internal Elements	External Elements	Adopting Activities (Warehouse Functions)	Adopting Activities (Warehouse Processes)
**The concordance coefficient (W)**	0.5000	0.2500	0.1375	0.5000
**A random size (** **χ** ** ^2^ ** **)**	10.0000	2.0000	1.6500	6.0000
**The lowest value of the concordance coefficient (W_min_)**	0.0805	0.0263	0.0487	0.0487

**Table 12 sensors-23-02213-t012:** The results of the ratio of the average rank to the total sum of ranks.

Elements of Expert Evaluation		Criterion Encryption Model					
**The ratio of the average rank to the total sum of ranks**	**Internal elements**	**Productivity**	**Morale and ease of work**	**Inventory**	**Order lead time**	**Operational productivity**	**Operational efficiency**
		0.1071	0.2262	0.1548	0.2619	0.1310	0.1190
	**External elements**	**Sale**	**Customer satisfaction**	**Competition**	**-**	**-**	**-**
		0.4167	0.2500	0.3333	-	-	-
	**Adopting activities (Warehouse functions)**	**Receiving**	**Put away**	**Storage**	**Packing and shipping**	**-**	**-**
		0.1892	0.2973	0.2432	0.2703	-	-
	**Adopting activities (Warehouse processes)**	**Forecasting**	**Inventory**	**Risk migitation**	**Monitoring**	**-**	**-**
		0.3000	0.1500	0.2000	0.3500	-	-

**Table 13 sensors-23-02213-t013:** The results of the reverse size.

Elements of Expert Evaluation		Criterion Encryption Model					
**Reverse size**	**Internal elements**	**Productivity**	**Morale and ease of work**	**Inventory**	**Order lead time**	**Operational productivity**	**Operational efficiency**
		0.8929	0.7738	0.8452	0.7381	0.8690	0.8810
	**External elements**	**Sale**	**Customer satisfaction**	**Competition**	**-**	**-**	**-**
		0.5833	0.7500	0.6667	-	-	-
	**Adopting activities (Warehouse functions)**	**Receiving**	**Put away**	**Storage**	**Packing and shipping**	**-**	**-**
		0.8108	0.7027	0.7568	0.7297	-	-
	**Adopting activities (Warehouse processes)**	**Forecasting**	**Inventory**	**Risk migitation**	**Monitoring**	**-**	**-**
		0.7000	0.8500	0.8000	0.6500	-	-

**Table 14 sensors-23-02213-t014:** The results of the indicator significance.

Elements of Expert Evaluation		Criterion Encryption Model					
**Significance indicator**	**Internal elements**	**Productivity**	**Morale and ease of work**	**Inventory**	**Order lead time**	**Operational productivity**	**Operational efficiency**
		0.1786	0.1548	0.1690	0.1476	0.1738	0.1762
	**External elements**	**Sale**	**Customer satisfaction**	**Competition**	**-**	**-**	**-**
		0.2917	0.3750	0.3333	-	-	-
	**Adopting activities (Warehouse functions)**	**Receiving**	**Put away**	**Storage**	**Packing and shipping**	**-**	**-**
		0.2703	0.2342	0.2523	0.2432	-	-
	**Adopting activities (Warehouse processes)**	**Forecasting**	**Inventory**	**Risk migitation**	**Monitoring**	**-**	**-**
		0.2333	0.2833	0.2667	0.2167	-	-

**Table 15 sensors-23-02213-t015:** Expert judgment of criterion importance indicator.

Elements Of Expert Evaluation		Criterion Encryption Model					
**Criterion importance indicator**	**Internal elements**	**Productivity**	**Morale and ease of work**	**Inventory**	**Order lead time**	**Operational productivity**	**Operational efficiency**
		0.2262	0.1071	0.1786	0.0714	0.2024	0.2143
	**External elements**	**Sale**	**Customer satisfaction**	**Competition**	**-**	**-**	**-**
		0.2500	0.4167	0.3333	-	-	-
	**Adopting activities (Warehouse functions)**	**Receiving**	**Put away**	**Storage**	**Packing and shipping**	**-**	**-**
		0.3514	0.2432	0.2973	0.2703	-	-
	**Adopting activities (Warehouse processes)**	**Forecasting**	**Inventory**	**Risk migitation**	**Monitoring**	**-**	**-**
		0.2000	0.3500	0.3000	0.1500	-	-

## Data Availability

Not applicable.

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
