# Peer review of "Research on Impact of IoT on Warehouse Management"

_sensors, 2023, doi:10.3390/s23042213_

Round 1

Reviewer 1 Report

See attached file.

Author Response

Response to Reviewer 1 Comments

Dear Reviewer, Thank you for your comments and time for evaluating our article. They were useful in adjusting the article (All changes were done using the "Track Changes" function in Microsoft Word). Our responses to your comments are as follows:

Point 1: English. Please, carefully proofread the paper.

Response 1: The remark has been taken into account. Corrections made.

Point 2: Please, emphasise the fact that this paper is actually a literature review. Indeed, at line 1 ”Article” is written.

Response 2: Thank you for your comment. Please note that a scientific article is an article that presents the results of scientific research or a valuable analysis of previous results and has the necessary structural parts. Considering the fact that the article contains an analysis of the scientific sources of the current situation, a study were conducted, a model was created and its approval was presented, therefore we do not agree with your statement.

Point 3: Please, avoid the use of contractions and repetitions.

Response 3: Thank you for noticing. It is taken into account.

Point 4: References. Some of them are too old (i.e., 1, 2, 4, 9, 10, 13, 16, 18, 19, 20, 28, 29, 30, 36, 38, 39, 42, 43, 46, 48, 49, 52, 54, 55, 56, 57, 58, and 59). Please, consider to substitute them with similar ones published from 2016 and on; alternatively, provide reasons to maintain them. Moreover, substitute References 12 and 35. Indeed, they are Master Thesis, and they probably did not undergo a peer-review process.

Response 4: Thank you for your comment. We added the new literature, but we also left the one we examined, since the purpose of the analysis was to show what and how it was examined over a long period of time and that new articles also refer to the same sources we named. Regarding the comment on sources 12 and 35, we would like to draw attention to a few facts: Master's theses are recognized as first-level (stage) research work, which is prepared together with a research supervisor who has at least a doctoral degree, and they are reviewed by at least 2 reviewers. Also taking into account the problematic issues of data protection and plagiarism, these mentioned sources must remain.

Point 5: Some crucial details about the research methodology for the literature review are missing. In particular, list the consulted databases and the keywords the Authors looked for.

Response 5: Thank you for your feedback. In our opinion, this would be redundant and inappropriate information in our article, and would not create additional added value according to our vision and presentation of information. In response to your comment-request: the search was carried out in the databases of various scientific parties according to the keywords: warehouses, warehousing, warehouse management, new approach to warehouse management, smart warehouses, Logistics 4, Industry 4, Supply chain and Internet of Things, Internet of Things and storage etc.

Point 6: Monitoring and tracking of objects in warehouses. Such problem is not properly covered in the paper. Indeed, it can be solved by making use of several enabling technologies of IoT falling within the context of LPWANs. In particular, objects can be given with sensor nodes periodically transmitting their state and position exploiting wireless protocols. Although issues related on transmission performances within environments like warehouses may arises, some LPWAN proved to be effective enough. To this end, please consider to include also 1, 2, 3, 4], but I strongly encourage the Authors to perform additional research.

Response 6: Thanks for the comment and suggestion. We have included the specified sources, but we will not conduct additional research within the framework of this article, since the primary goal was to Conduct a study of the impact of IoT on warehouse management based on company size due to the possibility of doing the conceptual model. We plan to carry out your proposed research in the further stages of the planned research.

Point 7: Table 1. The field ”Outbound applications” for the second row is missing.

Response 7: In this table (Prime Primary technologies of IoT technologies), based on the indicated sources, we made a generalization that will be relevant for the further model. In view of this, we will argue that "Outbound applications" for Communication technologies is too vague and cannot be named specifically, as in the previous section. Therefore, we corrected the technical error and wrote "-".

Point 8: In order to enhance the paper readability, consider to use bulleted points (e.g., lines 297-335, 524-539, 726-784, 835-850, 921-940).

Response 8: Thank you for your note - we have resolved this technical issue.

Point 9: Please, add a paragraph properly describing what is reported in Table 3.

Response 9: After fixing the above mentioned remark, this remark is no longer relevant in our view.

Point 10: Lines 345-346. Please, explain all the terms of the equation, because only some of them are reported in Table 4.

Response 10: The formula is removed and replaced with text information.

Point 11: Section 2.2. Please, expand the discussion.

Response 11: This is a generalization of the previous text, so we deleted subsection 2.2 and added the previous text to the literature review.

Point 12: Section 2.3, 2.4. Please, better contextualise them because, at the moment, they seem to be vaguely introduced.

Response 12: We have added information.

Point 13: Line 520. Please, list the “additional questions”.

Response 13: Given the fact that the questions has been presented both in textual expression and in the form of tables and graphs, we do not provide them, as this would be redundant information and unnecessarily burden the reader.

Point 14: Section 4 seems to be missing.

Response 14: Thank you for your feedback. This was a technical layout error.

Reviewer 2 Report

Based on the information provided, there does not appear to be any logical contradiction or inconsistency in the paper. The authors present a clear argument for the potential benefits of implementing the Internet of Things (IoT) in warehouse management, including improvements in real-time inventory accuracy, predictive maintenance, operational efficiency, work quality, and productivity, as well as additional benefits such as energy saving, improved sales, and customer satisfaction. The authors also acknowledge that the implementation of the IoT in warehouses can be costly, including installation costs, data breach costs, and issues with data privacy and governance, and suggest that companies carefully consider these costs before making a decision to implement the IoT. The authors present a model for the adoption of the IoT in warehouse companies, which takes into account the costs and benefits of the IoT, as well as the size of the company. Overall, the authors present a coherent and logical argument for the potential benefits and costs of implementing the IoT in warehouse management.

There are a few additional pieces of information that the authors could include in the paper to make it more suitable for an SCI academic review article. Some suggestions include:

A more detailed and systematic review of the existing literature on the use of the Internet of Things (IoT) in warehouse management. This could include a discussion of the various methods and approaches used in previous studies, as well as the main findings and limitations of these studies.

A more in-depth analysis of the data collected from expert interviews, including a discussion of the sampling strategy, the number of experts interviewed, and the types of questions asked. The authors could also include more detailed results and findings from the expert interviews, including any patterns or trends that emerged.

A more detailed discussion of the proposed model for the adoption of the IoT in warehouse companies, including a clear explanation of the variables and metrics used in the model, as well as the assumptions and limitations of the model.

A more detailed discussion of the implications of the study's findings for practice and future research, including specific recommendations for practitioners and researchers working in the field of warehouse management and the IoT.

More detailed information on the study's limitations and potential sources of bias, including a discussion of any potential confounding variables that may have impacted the study's results.

More detailed information on the study's contributions to the field, including a discussion of how the study builds on or adds to the existing literature on the use of the IoT in warehouse management.

It could be interesting for readers if the study included a discussion of the use of mobile IoT devices, such as autonomous guided vehicles (AGVs), drones, and personal devices, in warehouse management. These types of devices have the potential to significantly impact the efficiency and effectiveness of warehouse operations, and a discussion of their use and potential benefits in this context could be valuable for readers. Additionally, examining the challenges and limitations of using these types of mobile devices in warehouse environments could provide valuable insights for practitioners and researchers working in this field. Overall, including a discussion of the use of mobile IoT devices in warehouse management could add a new and interesting dimension to the study, and could potentially make it more relevant and valuable for readers. 

Authors should take advantage of the special issues targeting logistics and warehouse. https://www.mdpi.com/journal/drones/special_issues/5663HCFN3X

Author Response

Response to Reviewer 2 Comments

Dear Reviewer, Thank you for your comments and time for evaluating our article. They were useful in adjusting the article (All changes were done using the "Track Changes" function in Microsoft Word). Our responses to your comments are as follows:

Point 1: A more detailed and systematic review of the existing literature on the use of the Internet of Things (IoT) in warehouse management. This could include a discussion of the various methods and approaches used in previous studies, as well as the main findings and limitations of these studies.

Response 1: Thank you for your comment, but in our opinion, the information you requested is presented in Tables 1-3 (and at most Table 2 is focused on all generalizations).

Point 2: A more in-depth analysis of the data collected from expert interviews, including a discussion of the sampling strategy, the number of experts interviewed, and the types of questions asked. The authors could also include more detailed results and findings from the expert interviews, including any patterns or trends that emerged.

Response 2: Thank you for your comment. we expanded the methodology. We don't see the point in asking questions because it doesn't add value.

Point 3: A more detailed discussion of the proposed model for the adoption of the IoT in warehouse companies, including a clear explanation of the variables and metrics used in the model, as well as the assumptions and limitations of the model.

Response 3: Thank you for comment. In our opinion, there is enough information related to the description of the principle of operation of the model (the text has been technically corrected for clarity). We also want to point out that the model was compiled based on the review of scientific literature sources and the results of the conducted research, and the limitations of the research are indicated in the conclusions section.

Point 4: A more detailed discussion of the implications of the study's findings for practice and future research, including specific recommendations for practitioners and researchers working in the field of warehouse management and the IoT.

Response 4: Further research directions are indicated in the conclusions section. also, in the future, we can think about the possibilities of researching the application of drones, etc. but here, research of a different profile is needed.

Point 5: More detailed information on the study's limitations and potential sources of bias, including a discussion of any potential confounding variables that may have impacted the study's results.

Response 5: Thanks for the advice, the information has been added.

Point 6: More detailed information on the study's contributions to the field, including a discussion of how the study builds on or adds to the existing literature on the use of the IoT in warehouse management.

Response 6: In our opinion, the information presented is sufficient to show what is not analyzed in scientific sources. “Problems of scientific literature and gap. The empirical results give a good overview of IoT technologies which are used in the different warehousing companies. The IoT technologies can be implemented and can have a positive effect on warehouse functioning. Nevertheless, disadvantages such as financial aspects of data protection and safety need to be considered. The IoT technologies can be used in all warehousing operations. From the extensive literature review presented above, this research identified the following main research gaps: First, the existing researches so far mainly document the positive effect of IoT on companies' performance in general and warehouse management in particular. However, none of these studies take into account the associated side-effects of using IoT. That means, even though, using IoT might improve storage management and sales, the overall effect might not be positive. The overall effect of adopting IoT depends on the costs and benefits that IoT could bring to the company. While conducting a cost-benefit analysis is very critical to fully understand the effect of IoT on the overall performance of a company, no research was conducted (as to the knowledge of the researcher) that aimed at evaluating the general effect of IoT. We can consider the researches so far as partial analysis. For instance, using IoT might require a skilled labor force which increases the expense of the enterprise. Second, previous studies do not also differentiate the effect of IoT on different sizes of companies. The effect of IoT on firm performance could be context-specific. Thus, without considering such kind of disparities like company size suggesting the positive role of IoT on companies might be misleading. Therefore, in this research, the researcher will fill the aforementioned gaps by conducting a cost-benefit analysis of adopting IoT taking into account company size.” Taking into account these facts and our research, a conceptual model is created, the suitability of which is evaluated by experts. Therefore, in our opinion, we respond to your comment.

Point 7: It could be interesting for readers if the study included a discussion of the use of mobile IoT devices, such as autonomous guided vehicles (AGVs), drones, and personal devices, in warehouse management. These types of devices have the potential to significantly impact the efficiency and effectiveness of warehouse operations, and a discussion of their use and potential benefits in this context could be valuable for readers. Additionally, examining the challenges and limitations of using these types of mobile devices in warehouse environments could provide valuable insights for practitioners and researchers working in this field. Overall, including a discussion of the use of mobile IoT devices in warehouse management could add a new and interesting dimension to the study, and could potentially make it more relevant and valuable for readers.

Response 7: We agree with your expressed thoughts and thank you for your suggestions (and also the literature you suggest), they are really very valuable and worthy of attention in our further research (which we plan to take into account). Because the purpose of this article was to create a conceptual model for which, from our point of view, the results of the conducted research are sufficient.

Round 2

Reviewer 1 Report

The paper notably improved after its revision. However, there is still a comment of the last review round which seems not to be addressed. In particular, I am referring to the one on LPWANs. For the sake of convenience, I copy paste it below.

  • Monitoring and tracking of objects in warehouses. Such problem is not properly covered in the paper. Indeed, it can be solved by making use of several enabling technologies of IoT falling within the context of LPWANs. In particular, objects can be given with sensor nodes periodically transmitting their state and position exploiting wireless protocols. Although issues related on transmission performances within environments like warehouses may arises, some LPWAN proved to be effective enough. To this end, please consider to include also [1, 2, 3, 4], but I strongly encourage the Authors to perform additional research.

  1. [1]  Pietro Manzoni, Carlos T Calafate, Juan-Carlos Cano, and Enrique Hern ́andez- Orallo. “Indoor vehicles geolocalization using LoRaWAN”. In: Future Internet 11.6 (2019), p. 124.

  2. [2]  Gabriele Di Renzone, Ada Fort, Marco Mugnaini, Stefano Parrino, Giacomo Peruzzi, and Alessandro Pozzebon. “Interoperability among sub-GHz technologies for metallic assets tracking and monitoring”. In: 2020 IEEE International Workshop on Metrology for Industry 4.0 & IoT. IEEE. 2020, pp. 131–136.

  3. [3]  S Mangal and Ketan Rajawat. “Performance evaluation of LoRaWAN with decreasing RSSI values”. In: Int. J. Appl. Eng. Res. 14.2 (2019), pp. 177–182.

  4. [4]  Che Cameron, Wasif Naeem, and Kang Li. “Functional Qos metric for lorawan ap- plications in challenging industrial environment”. In: 2020 16th IEEE International Conference on Factory Communication Systems (WFCS). IEEE. 2020, pp. 1–6.

Reviewer 2 Report

1- The scientific contribution of the article has not been clearly explained. Please highlight and explain it in a bullet point format in the introduction section.

2- The article should attract the reader's interest. It continues for a long time in a plain writing style. The number of practical comparison tables should be increased and new ones should be added.

3- There are paragraphs consisting of only one sentence. These are writing errors.

4- Some tables are unreadable, please correct them.

5- Additionally, the objectivity of the applied method is very weak from a scientific point of view. The methodology and findings section is insufficient in its current form. The article is properly written as a review, but it is not mature enough as a research article.
